# Dynamic structure of active sites in ceria-supported Pt catalysts for the water gas shift reaction

Yuanyuan Li [1✉], Matthew Kottwitz[2], Joshua L. Vincent[3], Michael J. Enright[2], Zongyuan Liu[4], Lihua Zhang[5], Jiahao Huang [1], Sanjaya D. Senanayake[4], Wei-Chang D. Yang [6,7], Peter A. Crozier[3], Ralph G. Nuzzo[2,8] & Anatoly I. Frenkel [1,4]

Oxide-supported noble metal catalysts have been extensively studied for decades for the water gas shift (WGS) reaction, a catalytic transformation central to a host of large volume processes that variously utilize or produce hydrogen. There remains considerable uncertainty as to how the specific features of the active metal-support interfacial bonding—perhaps most importantly the temporal dynamic changes occurring therein—serve to enable high activity and selectivity. Here we report the dynamic characteristics of a $Pt/CeO_2$ system at the atomic level for the WGS reaction and specifically reveal the synergistic effects of metal-support bonding at the perimeter region. We find that the perimeter $Pt^0 - O$ vacancy$-Ce^{3+}$ sites are formed in the active structure, transformed at working temperatures and their appearance regulates the adsorbate behaviors. We find that the dynamic nature of this site is a key mechanistic step for the WGS reaction.

[1] Department of Materials Science and Chemical Engineering, Stony Brook University, Stony Brook, NY 11794, USA. [2] Department of Chemistry, University of Illinois, Urbana, IL 61801, USA. [3] School for Engineering of Matter, Transport and Energy, Arizona State University, Tempe, AZ 85287-6106, USA. [4] Chemistry Division, Brookhaven National Laboratory, Upton, NY 11973, USA. [5] Center for Functional Nanomaterials, Brookhaven National Laboratory, Upton, NY 11973, USA. [6] Physical Measurement Laboratory, National Institute of Standards and Technology, Gaithersburg, MD 20899, USA. [7] Institute for Research in Electronics and Applied Physics & Maryland NanoCenter, University of Maryland, College Park, MD 20742, USA. [8] Surface and Corrosion Science, School of Engineering Sciences in Chemistry, Biotechnology and Health, KTH Royal Institute of Technology, Drottning Kristinasväg 51, 100 44 Stockholm, Sweden. ✉email: Yuanyuan.li@stonybrook.edu

The WGS reaction ($CO + H_2O \rightarrow CO_2 + H_2$) is an essential processing step in the industrial production of synthesis gas—one used both to purify and to adjust the $H_2$ composition in important downstream processes such as the Fischer–Tropsch synthesis of hydrocarbons ($CO:H_2 = n:(2n + 1)$), methanol synthesis ($CO:H_2 = 1:2$), and the Haber–Bosch process for the production of ammonia[1]. Among all WGS catalysts, ceria-supported noble metal (Pt and Au) catalysts have been heavily investigated recently because they have excellent low-temperature activity, selectivity, and stability, and they are non-pyrophoric (an important feature for such uses as in polymer electrolyte membrane (PEM) fuel cells)[2,3].

The identity of the active species/sites of ceria-supported Au or Pt catalysts and the roles of the supports in the WGS reaction have been sources of intense debate. For example, several groups reported that the nonmetallic gold or platinum species/sites strongly associated with the support are responsible for the catalytic activity with the metal of the nanoparticles acting as spectator species[4–7]. Ding et al., however, observed that only metallic Pt nanoparticles show activity and that nonmetallic Pt single atoms ($Pt^{\delta+}$) are spectators on a variety of supports[8]. In terms of the support effects, Zhai et al. reported that a support only serves to stabilize the $Pt^{\delta+}$ sites[9]. Mechanistic/kinetic studies, based on advanced transient isotopic techniques, on the other hand, suggest that catalyst support provides active sites within a region (reactive zone) around the Pt nanoparticles[10]. Supports, especially oxygen vacancies of the supports, play an essential role in the activity of the catalyst because water dissociation, involved in two main mechanisms (the regenerative/redox and the associative mechanisms), is believed to occur at oxygen vacancy sites[11–13]. Vecchietti et al. introduced $Ga^{3+}$ to ceria supports to increase the concentration of surface oxygen vacancies. The catalytic activity of $Pt/Ce_{80}Ga_{20}$, however, was lower than that of $Pt/CeO_2$[12]. Ricote et al. added $Zr^{4+}$ to the ceria support. Although the zirconia doping did not increase oxygen vacancies, it enhanced oxygen surface mobility and increased the reaction rate[14].

These seemingly conflicting results imply that some important features of the active site that determine the catalytic reactivity are still missing in our current understanding and demonstrate the need for further understanding of all parts of this complex interacting system (metal catalyst, support, adsorbates) at the atomic level—information that is difficult to obtain without correlative, multimodal, in-situ/operando modes of characterization. In this work, we describe the correlative results of such a study. Within this work, combined methods of ex situ and in situ characterization have elucidated the unique electronic structure and bonding environment of the active sites (perimeter $Pt^0 - O$ vacancy$-Ce^{3+}$ sites) of $Pt/CeO_2$. The data demonstrate that strong dynamical forms of complexity exist in this system, features that convolve strong mediating perturbations due to adsorbates and metal center (Pt and Ce) valence- and bonding-state speciations.

## Results

**Size evolution of Pt species under working conditions.** An activity test (Fig. 1a) revealed the catalyst first starts to show activity at 180 °C. As the temperature is raised, the production of $H_2$ and $CO_2$ continues to increase. The catalyst is not active below 180 °C when the temperature ramp is reversed. To understand the effect of WGS reaction conditions upon the structure of the catalyst, the nature of the as-prepared and reacted (collected after the activity test) $Pt/CeO_2$ were evaluated using scanning transmission electron microscopy (STEM) and X-ray absorption spectroscopy (XAS). High-angle annular dark field (HAADF)-

STEM images of the as-prepared and reacted catalyst are shown in Fig. 1b, c, respectively. The Pt species in the as-prepared catalyst are not easily identified due to their ultra-small size (likely single-atom, vide infra) and highly dispersed nature, as seen by the patches of diffuse contrast in Fig. 1b. We note here that XAS spectra of the as-prepared catalyst show very similar features to those expected for ceria-supported Pt(II) single atoms (Supplementary Fig. 1)[15]. Over the course of the reaction, the Pt species aggregate into larger Pt nanoclusters (about 1.7 nm in diameter, Supplementary Fig. 2) which are seen clearly in Fig. 1c. The XAS measurements that are discussed later in the manuscript agree quantitatively with the STEM particle size measurements. These results, together with those obtained from in situ diffuse reflectance infrared Fourier-Transform spectroscopy (DRIFTS) measurements (Supplementary Fig. 3) and XAS measurements (Supplementary Fig. 4), suggest that during the activating temperature ramp, single Pt atom species in the as-prepared sample coarsen to metallic Pt nanoparticles at T ≥ 180 °C. These metallic Pt nanoparticles remain the predominant species upon ramp-down to RT. When correlated with activity results, the data demonstrate that Pt nanoclusters likely exist within the working temperature range of the catalyst (180–300 °C). We hereafter focus on the attributes of atomic structure presented by Pt nanoclusters in situ within this temperature range.

**Dynamics and heterogeneity of atoms in the active Pt species.** First, we performed in situ TEM measurements to observe the atoms within active Pt nanoparticles. A typical Pt nanocluster is shown first in CO (Fig. 2a1–a6, Supplementary Movie 1) and then in the WGS reaction environment (Fig. 2b1–b6, Supplementary Movie 2). Due to the weakening of the Pt–Pt bonds from the strong repulsive interactions occurring between adsorbed CO molecules and as a result of the elevated temperature[16,17], the Pt atoms are generally quite dynamic under CO at 200 °C, except notably for those atoms deep in the nanocluster core and at/close to the interface between the nanocluster and the support. In contrast, under the WGS condition wherein water is introduced, almost all the outer Pt atoms become more stabilized/localized except for the ones located at the perimeter sites of the cluster (as shown by the yellow arrows in Fig. 2b3–b5). At the perimeter sites, the Pt atomic columns are seen to disappear (Fig. 2b3) and reappear (Fig. 2b4) in sequential images, indicating that they remain dynamically mobile. These observations suggest that the atoms at different sites within the nanocluster dynamically and heterogeneously change under in situ conditions. The dynamics of the Pt atoms at the perimeter, in comparison to the relatively static Pt atoms at the terrace/step surface sites, demonstrate their uniqueness as specific metal centers, ones where chemical activities are likely occurring (vide infra).

In the present data, we further observe a distortion in the interplanar spacing (plane bending) that is present at the surface of the nanocluster, here visible in Fig. 2b2 and indicated with dashed yellow lines for clarity. The observed bending is an indication that the surface atoms are not in a bulk terminated configuration and that anisotropic/localized bond strains—ones subject to perturbations due to both adsorbates and specific support bonding interactions—are important features of their structure. We will now focus on the dynamic interactions of the perimeter sites with adsorbates and support atoms in order to better understand the observed phenomena.

**In situ variation of adsorbate bonding at perimeter Pt sites.** To provide additional insights into the bonding of adsorbates with specific Pt sites, DRIFTS measurements were performed first under CO and then WGS conditions. Under CO (Fig. 3a) at high

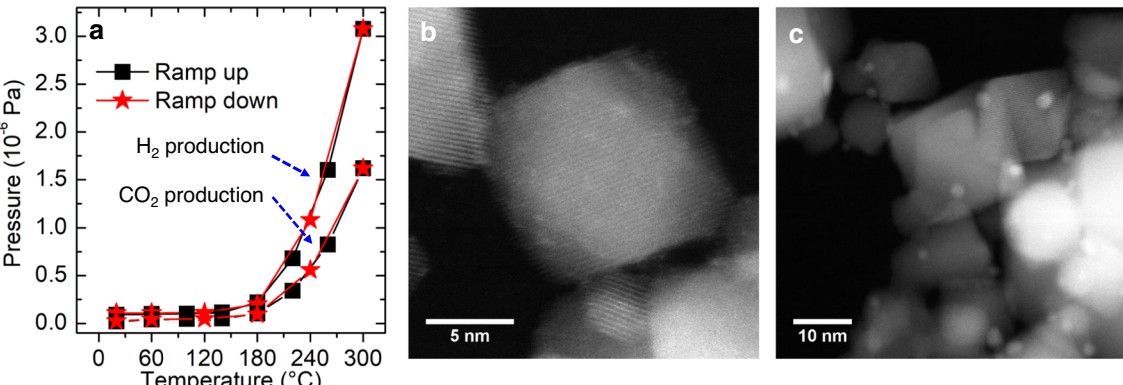

**Fig. 1 Activity and the catalyst before and after the activity test. a** Activity test: the $H_2$ and $CO_2$ productions at different temperatures in the ramp-up and down processes. **b** The STEM annular dark field (ADF) image of the as-prepared Pt/ceria catalysts. **c** The STEM-ADF image of the reacted Pt/ceria catalyst.

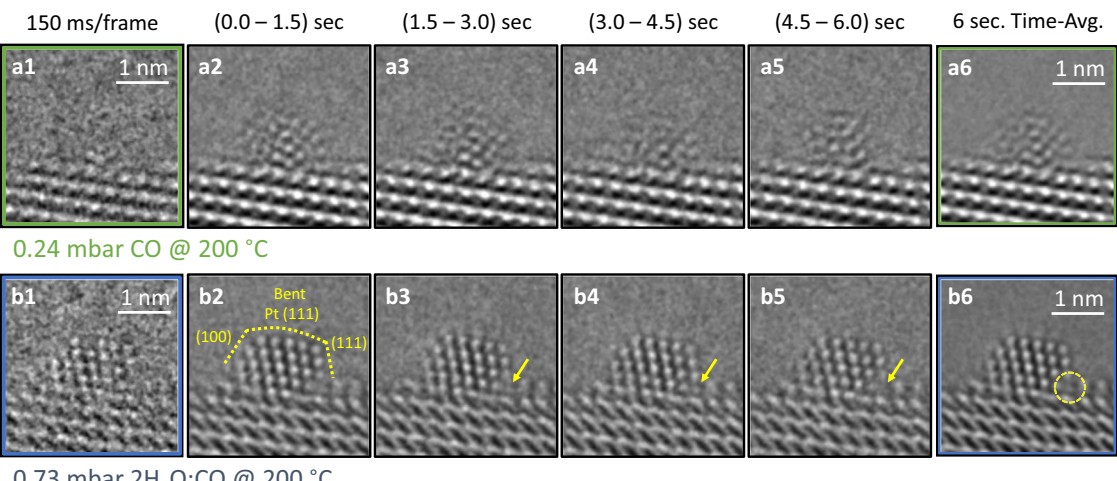

**Fig. 2 Dynamic structural response and behavior of the Pt/CeO₂ catalyst. a1–a6** Images of the Pt/CeO₂ catalyst at 200 °C in CO gas. **b1–b6** Images of the Pt/CeO₂ catalyst at 200 °C in WGS reaction conditions. **a1/b1** shows the catalyst (close to the mean size of 1.7 nm) in a single 150 millisecond frame (ms/frame). Time-averaged image series from sequential 1.5 s intervals are shown in Figs. a2–a5 and b2–b5. The CeO₂-supported Pt nanoparticle adopts a bent, facetted morphology in the WGS condition, which is highlighted by the dashed outline and facet labels. Dynamic structural fluctuations are seen at perimeter sites as indicated by the yellow arrow. **a6/b6** show the time-averaged image over the entire 6 s movie for each condition. The corresponding in situ TEM movies can be found in Supplementary Movie 1 and 2. The same Pt nanoparticle is shown in all frames. The behavior exhibited by the Pt nanoparticle in Fig. 2 is typical of the many observed during the in situ TEM experiment. Additional images from a different particle, for example, are shown in Supplementary Fig. 5.

temperatures (300 °C, 240 °C, and 180 °C), infrared bands of high frequency (HF) and low frequency (LF) atop/bridge bound CO on Pt atoms are observed; at lower temperatures (120 °C and RT), the LF bands disappear. The LF bands are associated with CO bonded to low coordinated Pt atoms in proximal contact with reduced $Ce^{3+}$ sites along the metal−support interface ($Pt^0 − O$ vacancy$−Ce^{3+}$)[18–24]. The appearance of the LF bands at high temperatures, therefore, suggests that a correlation exists between Pt atoms proximal to the catalyst perimeter and reduced ceria sites.

As shown in Fig. 3b–f, the introduction of $H_2O$ does not affect the line shapes of these modes but does influence their intensities. At low temperatures (26 °C and 120 °C), an increase of the HF peak intensity, together with a redshift, signifies an interaction between water molecules, Pt, and CO via one of several possible mechanisms noted in prior literature: (i) partial charge transfer to the CO $2\pi^*$ orbital due to the modification of the metal particle potential by the lone pair of a bound water molecule; or (ii) dipole-dipole interactions between co-adsorbed CO and water

molecules[25–30]. Interestingly, this trend is reversed at higher temperatures (at which the dynamic perimeter $Pt^0 − O$ vacancy $−Ce^{3+}$ sites appear). Upon further addition of $H_2O$ to CO at 180 °C, 240 °C, and 300 °C, the intensities of the HF peaks decrease while those of the LF peaks increase, an indication of the migration of some CO adsorbates from high to low coordination perimeter Pt sites due to the change of gas condition from CO to WGS. Such changes seem stronger with the decrease of temperature from 300 °C to 180 °C which could be due to the temperature effect on the CO coverage[27,31].

In summary of the TEM and DRIFTS observations, we conclude that the perimeter $Pt^0 − O$ vacancy$−Ce^{3+}$ site responds dynamically to the temperature and that CO (one of the two reactants) migrates to it at high temperature. We, therefore, hypothesize that it is the active site for the WGS reaction. To further investigate this hypothesis, we turn our attention to another reactant, $H_2O$.

Water dissociates into hydroxyl and atomic H species on the surface of ceria supports. The hydroxyl group bound to the

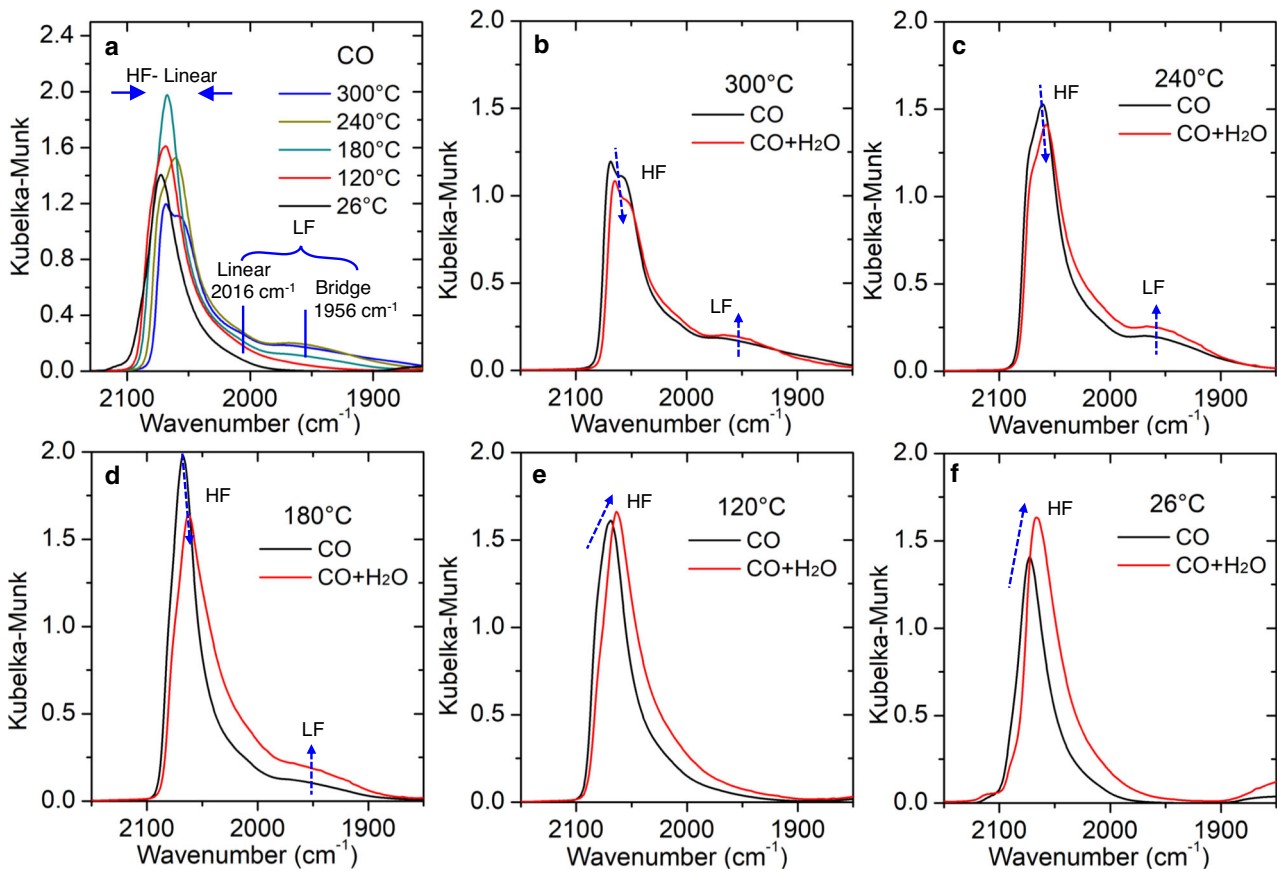

**Fig. 3 DRIFTS studies of CO bonds on Pt nanoclusters under CO and WGS conditions.** The experiments were performed by decreasing the temperature from high to low. **a** The temperature-dependent CO – DRIFTS spectra of Pt/CeO₂ catalyst collected in the ramp-down process. Bands in high frequency (HF) region (2072–2054 cm⁻¹) are assigned to linearly adsorbed CO on terrace or step Pt sites[18,22,53,54]. In low frequency (LF) region, the band centered at ~2016 cm⁻¹ is assigned to CO linearly adsorbed on perimeter Pt sites while the band centered at ~1956 cm⁻¹ is assigned to a bridged CO adsorption at the metal support interface[18-24]. **b–f** At 300 °C, 240 °C, 180 °C, 120 °C, and 26 °C, the changes of CO bands on the Pt surface sites upon the change of condition from CO to WGS. The positions of HF peaks were labeled in Supplementary Fig. 6.

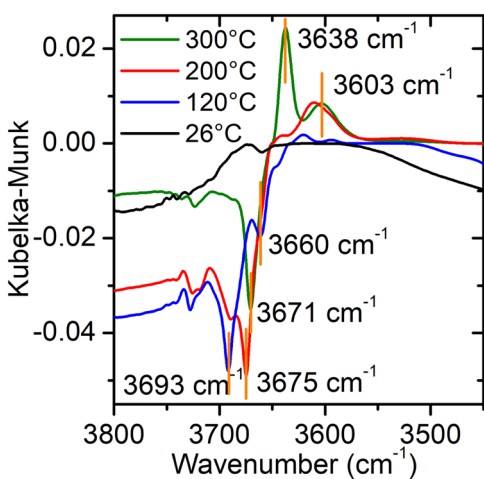

**Fig. 4 DRIFTS studies of hydroxyl species on the surface of ceria support.** The figure shows the changes of pre-adsorbed OH bands after CO was introduced at 300 °C, 200 °C, 120 °C, and 26 °C (see Methods for experimental details). Band assignments are based on previous reports:[32,33] 3693 cm⁻¹ — mono-coordinated OH (Type I) on Ce; 3675 cm⁻¹, 3671 cm⁻¹, and 3660 cm⁻¹ — doubly bridging OH (Type II-A) on Ce; 3638 cm⁻¹ — doubly bridging OH on Ce with oxygen vacancy in the vicinity (type II-B); 3603 cm⁻¹ — triply bridging OH (Type III) on Ce.

surface of ceria is easily detected by DRIFTS measurements. Figure 4 shows difference spectra illustrating the changes that occur in the bonding of hydroxyl moieties that are pre-adsorbed on ceria in the presence of CO at 300, 200, 120, and 26 °C. At both 300 °C and 200 °C, the difference spectra show complex negative and positive features, indicating that a redistribution of –OH bonding states on the ceria occurs after the introduction of CO. The increase in the intensity of the type II-B OH band (at 3638 cm⁻¹) is an indication that the ceria support has been reduced, likely from surface oxygen elimination, and it suggests the association of OH bonds with surface reduced ceria sites at working temperatures[32,33]. Previous reports also suggest that the active –OH groups are classified as type-II –OH groups formed on partially reduced ceria and reside within a narrow zone around the periphery of Pt–ceria interface[34].

Combining these results (Fig. 4) with the insights previously discussed (Fig. 3), we hypothesize that the two reactants of the WGS process (CO and H₂O) must associate in part with proximal contacts occurring at the perimeter $Pt^0 - O$ vacancy$-Ce^{3+}$ sites at working temperatures. To verify this point, it is necessary to demonstrate that the reduced ceria sites that adsorb the hydroxyl groups are in fact located near the perimeter Pt sites. A precise quantification of their fractional densities can establish the existence of such a correlation between the number of perimeter $Pt^0$ sites and the number of oxygen vacancies/$Ce^{3+}$ on the ceria surface at working temperatures.

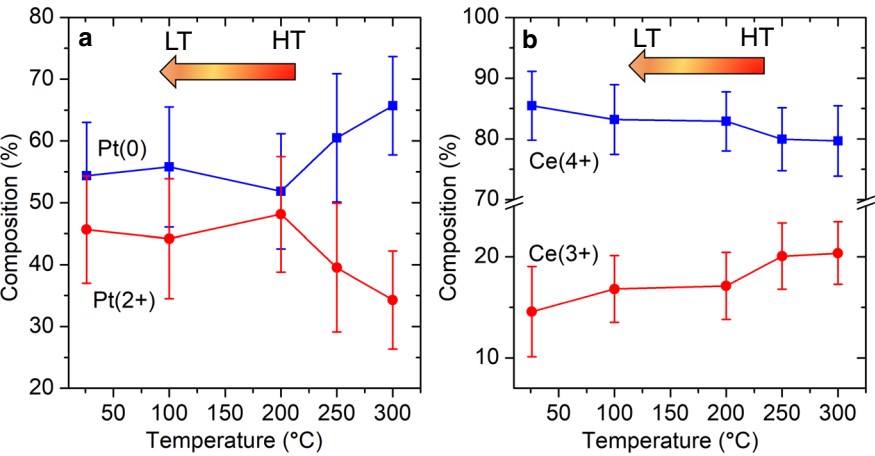

**Fig. 5 AP-XPS studies of Pt nanoclusters supported on ceria.** The valence state change of **a** Pt and **b** Ce under WGS conditions when the temperature drops from 300 °C (high temperature (HT)) to RT (low temperature (LT)). The standard deviation errors obtained from fitting peak areas are also shown. The corresponding Pt 4$f$ and Ce 3$d$ region XPS experimental and fitting spectra are shown in Supplementary Fig. 7.

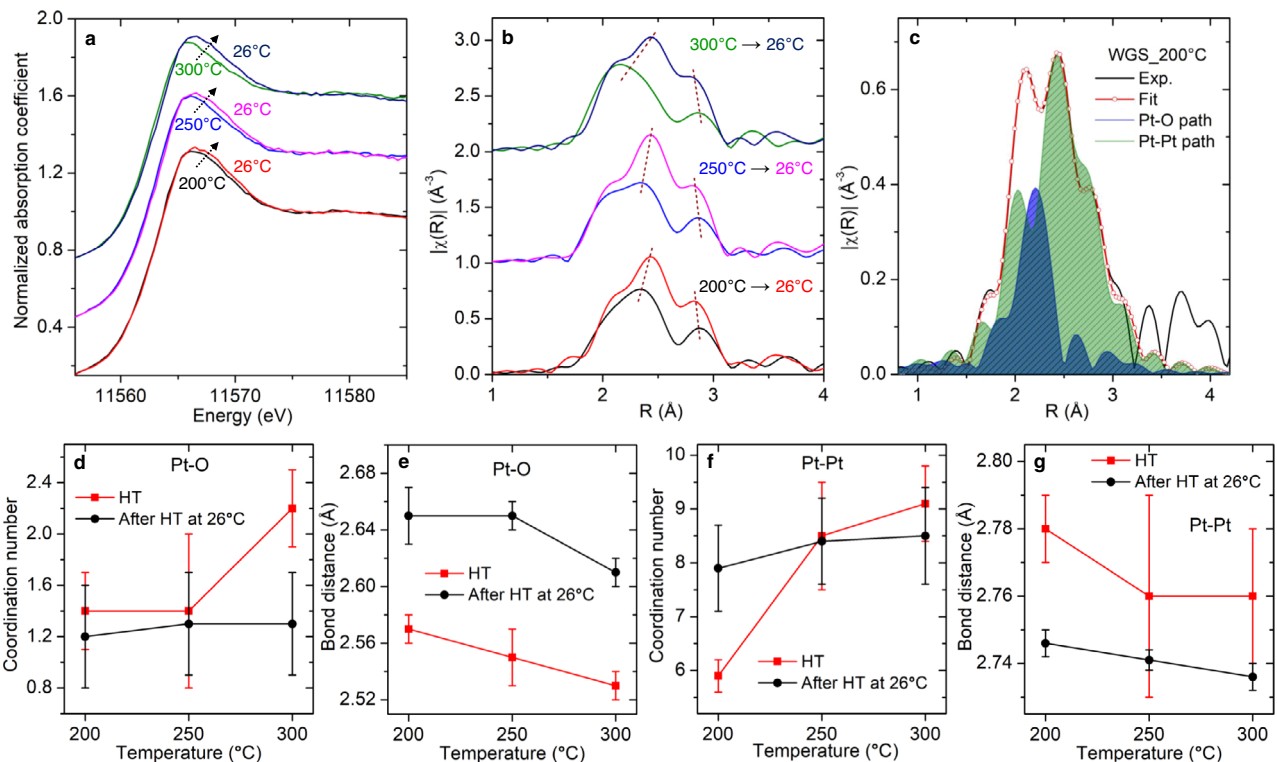

**Fig. 6 XAS studies of working Pt nanocluster/CeO$_2$ for WGS reaction. a** Normalized Pt L$_3$ XANES spectra and **b** Fourier transform magnitudes of $k^2\chi(k)$ EXAFS spectra of Pt/CeO$_2$ under WGS condition at 200 °C, 250 °C, and 300 °C. For comparison, the RT spectrum was also collected after each measurement at high temperatures. Note that the spectra, for 250 °C and 300 °C, are shifted vertically by 0.4 and 0.8, respectively, in (**a**) and by 1.0 and 2.0, respectively, in (**b**) for clarity. **c** The representative figure shows the agreement between the experimental spectrum and the fitted curve for the data collected in the WGS condition at 200 °C. The fitting model includes two paths: Pt-O and Pt-Pt. The contribution of Pt-O and Pt-Pt path is also plotted for reference. The temperature-dependent change of **d** Pt-O coordination number, **e** Pt-O bond distance, **f** Pt-Pt coordination number, and **g** Pt-Pt bond distance under WGS conditions (red). For comparison, the results obtained for the spectrum collected at RT after each high temperature (HT) are also included (black). The error bars were obtained by performing analysis with IFEFFIT package[55]. The best-fitting results are listed in Supplementary Table 2. The experimental and fitted spectra are compared in Supplementary Fig. 9.

**Active oxygen vacancies associate with perimeter Pt$^0$ sites**. To quantify the ratio of perimeter Pt$^0$/Ce$^{3+}$, we carried out AP-XPS measurements of the Pt 4$f$ and Ce 3$d$ regions under WGS conditions (Fig. 5 and Supplementary Fig. 7). Though XPS reveals the effective valence state of Pt averaged over the entire nanocluster, positively charged Pt$^{2+}$ atoms (Supplementary Fig. 8), are most likely located at the interface with the Ce oxide support[35].

Therefore, changes in Pt$^{2+}$ concentration indicates the nanocluster-support interface restructures when cooled. Generally, there is a higher fraction of Pt$^0$ than Pt$^{2+}$ at all temperatures. Starting from about 65% Pt$^0$ and 35% Pt$^{2+}$ at 300 °C, the fraction of Pt$^0$ decreases linearly as the temperature decreases to 200 °C, at which point the nanocluster contains approximately 55% Pt$^0$ and 45% Pt$^{2+}$. This ratio is retained through further

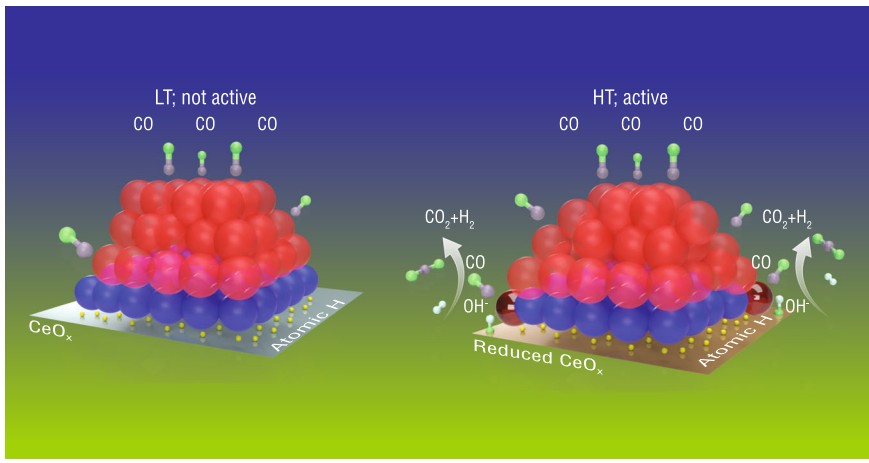

**Fig. 7 The schematics of the inactive and active structure of the Pt nanocluster/CeO₂.** At LT (low temperature (LT) < 180 °C), the catalyst is inactive while at HT (high temperature (HT): 180–300 °C), the catalyst is active. In the schematics, the red and blue spheres represent Pt atoms that are metallic (zero state) and oxidized (2+), respectively. For the active structure, the two maroon spheres represent active perimeter Pt sites.

cooling to RT. The numbers suggest that approximately 22% of interfacial Pt atoms are reduced to $Pt^0$ at 300 °C. Among the interfacial Pt atoms, those reducible atoms are most likely at the perimeter. They are the least coordinated and most affected by the support and adsorbates. Thus, approximately half of the perimeter Pt atoms are reduced to the zerovalent state. These high temperature $Pt^0$ perimeter sites strongly associate with reduced ceria sites as indicated by DRIFTS (the emergence of high-temperature perimeter $Pt^0 - O$ vacancy$-Ce^{3+}$ sites) and Ce $3d$ AP-XPS data. Under the WGS reaction, the concentration of $Ce^{3+}$ is 20% at 300 °C, which is 5% higher than that at RT. The rest of the Pt atoms at the interface remain in 2+ state and interact with the ceria support, thereby preventing further small cluster agglomeration.

AP-XPS evaluation of Pt $4f$ and Ce $3d$ reveal the relative concentrations of Pt and Ce to be approximately, a 1:9 ratio (Supplementary Table 1). Within the 1.7 nm Pt nanocluster, ~10% of Pt atoms are perimeter $Pt^0$ site and therefore represent about 1% of all metal (Pt and Ce) ions. At 300 °C, the concentration of $Ce^{3+}$ is about 20%. Assuming all $Ce^{3+}$ stay at the surface of the support[36], the total fraction of $Ce^{3+}$ is 18%, corresponding to 9% oxygen vacancies. Analogous estimates for the RT species suggest 7% oxygen vacancies (Fig. 5b and Supplementary Table 1). Assuming they are retained throughout the cooling process, the additional 2% oxygen vacancies correlate with the appearance of perimeter $Pt^0$ sites. Since perimeter $Pt^0$ sites associate with oxygen vacancies, this implies the increased oxygen vacancy sites appear near the perimeter $Pt^0$ sites at working temperatures. This reinforces our hypothesis that perimeter $Pt^0 - O$ vacancy$-Ce^{3+}$ sites bring the two reactants closer.

**Fluxional bond dynamics of Pt nanoclusters.** To provide a better understanding of the structural dynamics of ceria supported Pt nanoclusters, in situ XAS measurements were performed at Pt $L_3$ edge. As shown in Fig. 6a, b, the XAS spectra collected at working temperatures (200 °C, 250 °C, and 300 °C) show changes compared to those collected at corresponding post-reaction RT, suggesting that the catalyst is dynamic and adopts different structures when it is in its active catalytic state. The perturbations seen in these data illustrate underpinnings related to both atomic and electronic structural origins. At working temperatures, the white line (near 11,566 eV) is consistently shifted to lower energies and becomes narrower compared to RT.

The decreased white line intensity and the redshift suggest a reduction of Pt occurs at elevated temperatures, in agreement with XPS results.

A quantitative analysis was performed to elucidate the structural changes that are evidenced in the R-space EXAFS data (Supplementary Fig. 9). As shown in Fig. 6(d–g), the coordination numbers and the distances of Pt-O and Pt-Pt pairs change with temperature under WGS conditions. These changes in bonding are reversible and upon returning to RT revert to the initial resting state of the catalyst. At RT, the coordination number of the Pt-Pt pairs provided by the analysis is 8.3 ± 0.5 (Fig. 6f). This value corresponds closely to the expectations for an idealized, hemispherical cuboctahedral cluster with the size of approximately 1.7 nm[37]. We also found the bond length of Pt-O to be about 2.6 Å (Fig. 6e), significantly longer than the 2.0 Å to 2.2 Å range expected for direct Pt-O bonding interactions[38]. Long metal-oxygen distances (2.5 Å to 2.7 Å) have also been seen for supported Pt treated in $H_2$ at T < 350 °C as well as for clusters bound to partially-hydroxylated supports[38,39]. The long bond in each case was ascribed to the presence of an additional atom (i.e. atomic chemisorbed H) residing at the interface between the metal atoms and the support[39–42]. The adsorbed H atoms recombine into $H_2$ at working temperatures (Fig. 1a) and such recombination occurs with the appearance of active oxygen vacancies (near perimeter $Pt^0$ sites) which act as specific sites for $H_2$ activation[43].

As shown in Fig. 6e, at high temperatures, the bond distance of Pt-O is within the range from 2.53 Å to 2.57 Å, about 0.1 Å shorter than that at RTs (from 2.61 Å to 2.65 Å). Such change could be related to the reduced ceria support. The missing oxygen atoms in the ceria unit cell elongate the $Ce^{3+}$-O bond (~0.1 Å longer than that of $Ce^{4+}$-O)[44], which, via interface interaction, drives the structure modification of Pt clusters. In addition, a distortion in the lattice plane of Pt clusters was observed in electron microscopy results (Fig. 2b2).

**The catalytic activity at/near the perimeter $Pt^0$ sites.** The data described above make it possible to develop a schematic depiction of the atomic structure of the catalyst at low and high temperatures (Fig. 7), which helps to rationalize structure-activity correlations developed for the WGS reaction over Pt/CeO₂ catalysts. At low temperatures (< 180 °C), the Pt nanoclusters and the ceria support are inactive. Herein the reactant CO adsorbs on the step/terrace $Pt^0$ sites. The reactant $H_2O$, via dipole-dipole interaction

with CO, affects the CO-Pt bonding, and further likely dissociates into hydroxyl and atomic H: the hydroxyl groups bond to the surface of the ceria support, leaving atomic H species at the interface of the Pt nanocluster and the support. At elevated temperatures where the catalyst is active, the reduction of the ceria support and Pt nanoclusters produces new $Pt^0 - O$ vacancy $-Ce^{3+}$ sites localized at the perimeter of the Pt cluster. These changes in valence states and vacancy concentration at the nanocluster-support interface produce significant associated strains in the metal-metal bonds of the Pt clusters. The distorted structure may facilitate the facile transport of CO molecules to the perimeter $Pt^0$ surface-sites that comprise the active bonding ensemble for the catalytic reaction[45,46].

Though the associative mechanism may also play a role (Supplementary Fig. 10), the perimeter $Pt^0 - O$ vacancy $-Ce^{3+}$ sites are more likely associated with a regenerative redox mechanism. By using steady-state isotopic transient kinetic analysis (coupled with DRIFTS and mass spectrometry), Kalamaras et al. studied Pt/ceria catalysts of different sizes (1.3–8.0 nm). These results also suggest that at 300 °C, the WGS reaction proceeds largely via the redox mechanism with the reactive site located along the periphery of the Pt–CeO$_2$ interface (the specific rate based on the length of periphery of the Pt–ceria interface increases linearly with respect to particle size)[34]. In the redox mechanistic pathway, the hydroxyl groups donate atomic oxygen to the reduced support, which is subsequently re-reduced by donating the atomic oxygen to CO[34]. The oxidation of ceria at the perimeter of the nanocluster would cause the corresponding oxidation of the perimeter Pt atoms and strong bonding between the perimeter Pt atoms and the support. Regenerative redox mechanism requires oxidation state changes are dynamically reversible (reversible change of $Pt^0$ to $Pt^{2+}$ and $Ce^{3+}$ to $Ce^{4+}$). This is consistent with the dynamic restructuring of the perimeter atoms observed by TEM. Such repeating transitions in the electronic structure of the perimeter Pt atoms and their interaction with the support would markedly affect CO-Pt binding energy[47] and increase the mobility of perimeter Pt atoms as observed in in situ TEM. The higher mobility of the perimeter atoms has also been observed in ceria supported Au nanoparticles under CO oxidation[45], a reaction that most likely proceeds via a metal-assisted Mars–van Krevelen mechanism with oxygen vacancy exchange involved[48,49]. Here we note the importance of active oxygen vacancies. Oxygen vacancies also exist at low temperature but are not involved in the reaction. At high temperatures, only those newly emerged oxygen vacancies nearby the active Pt sites play important roles including adsorbing active hydroxyl groups, transporting oxygen atoms, and facilitating the recombination of H atoms which remain associated with sites at the nanocluster–support interface. The synergistic effects between O vacancy $-Ce^{3+}$ and active perimeter $Pt^0$ sites seem to best explain why only increasing the number of oxygen vacancies is not sufficient for improving the reaction rate[12,14], and why not all perimeter Pt sites are able to be involved in the reaction[50]. Experimentally, to improve the reactivity of Pt/ceria catalysts for the WGS reaction, the dopants should be able to increase the concentration or improve the activity of oxygen vacancies that involved in the perimeter $Pt^0 - O$ vacancy $-Ce^{3+}$ sites[51].

In this work, we studied a system of ceria supported Pt for the WGS reaction by combining multiple in situ techniques including DRIFTS, AP-XPS, XAS, and ETEM. With the multiple sources of information on Pt nanocluster/ceria combined, we revealed the complexity of the catalysts under working conditions. The complexity involves the structural evolution of the catalysts, heterogeneity and dynamic behavior of atoms located at different places of the nanostructure. In this work, the functional species are found to be ceria supported Pt nanoclusters, in which

multiple platinum states coexist. The WGS reaction proceeds most likely via the redox mechanistic path at the dynamic perimeter $Pt^0$-O vacancy-$Ce^{3+}$ site at which CO oxidation, water reduction, and H recombination occur. Accompanying the formation of the new perimeter $Pt^0$-O vacancy-$Ce^{3+}$ sites are a bonding environment change occurring at the interface where H atoms are stored and a localized/substantive intraparticle distortion which facilitates the CO transport to the perimeter $Pt^0$ sites.

## Methods

**Sample preparation.** To prepare Pt/ceria sample, 0.5 g of cerium(IV) oxide nanopowder (<25 nm) was dispersed in a solution consisting of 2.0 g of Urea and 8 mL of water. While stirring, 3.3 g of a solution (approximately 1 wt% Pt) of chloroplatinic acid hydrate in water was added to the dispersion. All chemicals were purchased from Sigma-Aldrich. Ultrapure water (18.2 MOhm) was provided by a Millipore purification system. The mixture was sealed in a vial and stirred for 24 h in an oil bath at 95 °C. Afterward, the postdeposition sample was separated and washed by 3 cycles of centrifuging/redispersing in water to remove the residual ions from the precursors. The sample was then dried overnight at 60 °C before being crushed into a powder and calcined at 500 °C (10 °C/min heating ramp). The resulting platinum loading of the sample was measured by Inductively Coupled Plasma (ICP) elemental analysis and determined to be 1.85 wt%. Note, certain commercial equipment, instruments, or materials are identified in this paper in order to specify the experimental procedure adequately. Such identification is not intended to imply recommendation or endorsement by the National Institute of Standards and Technology, nor is it intended to imply that the materials or equipment identified are necessarily the best available for the purpose.

**Activity test.** For the activity test, the as-prepared catalyst was loaded in a quartz tube (2 mm I.D., 2.4 mm O.D.) and mounted in a Clausen cell[52]. For the reaction, 1% CO (balanced with He; total flow rate of 25 ml/min) traveled through a water bubbler to carry water vapor (2.3%) to the sample. The activity test was executed by a stepwise increase of the reaction temperature from room temperature (RT) to 300 °C and returning to RT. The reactants and products were analyzed by a mass spectrometer (Hiden QGA).

**Electron microscopy.** Ex situ electron microscopy was carried out on a Hitachi 2700D scanning transmission electron microscope equipped with a probe aberration corrector operated at 200 kV. In situ environmental transmission electron microscopy (ETEM) experiments were conducted using an aberration-corrected FEI Titan ETEM operated at 300 kV. The third-order spherical aberration coefficient (C3) of the aberration corrector was tuned to a slightly negative value of approximately −15 μm, yielding a white-column contrast for the atomic columns in the resultant images. TEM samples were prepared by dispersing the Pt/CeO$_2$ powder onto a windowed micro electro-mechanical system (MEMS)-based Si$_3$N$_4$ chip, which was then loaded into a DENSsolutions Wildfire heating holder. After loading the sample into the ETEM, 24 Pa of CO gas was introduced into the environmental cell and the catalyst was reduced in situ at 300 °C for 2 hours. The catalyst was subsequently cooled to 200 °C and imaged under the same pressure of CO gas. To produce a water-gas shift reaction environment, at 200 °C, water vapor was introduced to the cell with a H$_2$O:CO reactant gas ratio of 2:1, resulting in a total cell pressure of 73 Pa. Images were acquired using a Gatan K2 IS direct electron detector operated at 20 frames per second with an electron flux of $9.6 \times 10^{13}$ A m$^{-2}$ and a total exposure time of 6 s. The electron beam was blanked when images were not being acquired. Static high-resolution images were recorded from many areas of the catalyst under CO and then WGS conditions. The time-resolved images were temporally binned by summing to various desired time resolutions (e.g., 0.15 s, 1.5 s, and 6.0 s), producing so-called time-averaged images with an increased signal-to-noise ratio.

**Diffuse reflectance infrared Fourier-transform spectroscopy (DRIFTS).** The spectra were collected using a Nicolet 6700 FTIR spectrometer equipped with a rapid-scanning liquid nitrogen-cooled mercury cadmium telluride (MCT) detector and a praying mantis high-temperature reaction chamber (Harrick Scientific Products). Using this reactor, the catalytic behavior of the catalyst was also tested and found to be very similar to that obtained using the Clausen reactor (Supplementary Fig. 11). Two types of temperature-dependent DRIFTS measurements were performed. In Measurement I (Fig. 3), at each temperature, the sample was treated in the following order: He (the background spectrum was collected) → 1% CO/He → 1% CO/2.3% water vapor/He. In Measurement II (Fig. 4), to study the surface OH bonding (3800 cm$^{-1}$ to 3500 cm$^{-1}$), at each temperature: 2.3% H$_2$O/ He → He (the background spectrum was collected) → 1% CO/He mixture. For both measurements, the temperature was increased stepwise from RT to 300 °C and back to RT. The total gas flow rate was 25 ml/min. Under each condition, the spectra were continuously collected until no further changes were observed. Each spectrum (256 scans) was collected with a resolution of 4 cm$^{-1}$.

**Ambient pressure X-ray photoelectron spectroscopy (AP-XPS).** A commercial SPECS AP-XPS instrument equipped with a PHOIBOS 150 EP MCD-9 analyzer at the Chemistry Division of Brookhaven National Laboratory (BNL) was used for XPS analysis (resolution: ~0.4 eV). A Mg K-alpha X-ray anode (1254 eV) was used for all measurements. Incident Mg K-alpha (1254 eV) incident X-rays have a maximum escape depth of about 6.2 nm, deep enough to probe the entire Pt nanocluster. The Ce $3d$ photoemission line with the strongest $Ce^{4+}$ feature (916.9 eV) was used for energy calibration. The powder sample (Pt/CeO$_2$ catalyst) was pressed on to an aluminum plate and loaded into the AP-XPS chamber at ultra-high vacuum. For the reaction, 1.3 Pa of CO + 3.0 Pa H$_2$O (CO:H$_2$O = 1:2.3) was introduced into the reaction chamber through a high precision leak valve and O $1s$, Ce $3d$, and Pt $4f$ regions were collected at different temperatures. The temperature was increased stepwise from RT to 300 °C and back to RT.

**X-ray absorption spectroscopy (XAS).** XAS measurements were performed at Beamline 20-ID-C of the Advanced Photon Source (APS), Argonne National Laboratory. The sample powders were loaded in a quartz tubing (2 mm I.D., 2.4 mm O.D.) and mounted in a Clausen cell. The Pt L$_3$ XAS data were collected in fluorescence mode by using a Vortex (Hitachi) silicon drift detector. For the in situ measurements, 1% CO (balanced with He; the total flowrate of 25 ml/min) carrying water vapor (2.3%) flows through the catalyst. The measurements were performed under WGS reaction conditions through a sequence of heating and subsequent cooling to RT steps (RT → 100 °C → RT → 150 °C → RT → 200 °C → RT → 250 °C → RT → 300 °C → RT). Once the desired temperature was reached, after about 15 min, the XAS data were collected.

## Data availability
The data supporting the findings of this study are available within the article and its Supplementary Information file. Additional information and figures are provided in the Supplementary Information file and Supplementary movies. Other data are available from the authors upon request.

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

## Acknowledgements

Y.L., A.I.F. and R.G.N. acknowledge support for this work by the U.S. Department of Energy, Office of Basic Energy Sciences under Grant No. DE-FG02-03ER15476. Reaction tests and DRIFTS measurements at Brookhaven National Laboratory's Chemistry Division and XAFS data analysis were made possible due to the Program Development fund 21-017 to A.I.F. M.K. and R.G.N. acknowledge fellowship support from the Department of Chemistry at the University of Illinois. Work by S.D.S. and Z.L. at Brookhaven National Laboratory was supported by the US Department of Energy, Office of Basic Energy Sciences, Division of Chemical Sciences, Geosciences and Biosciences under contract no. DE-SC0012704. S.D.S. is partially supported by a US Department of Energy Early Career Award. This research used Hitachi2700C STEM of the Center for Functional Nanomaterials, which is a U.S. DOE Office of Science Facility, at Brookhaven National Laboratory under Contract No. DE-SC0012704. The XAFS experiments used resources of the Advanced Photon Source Beamline 20ID at Argonne National Laboratory, which is an Office of Science User Facility operated for the U.S. Department of Energy (DOE) Office of Science and was supported by the U.S. DOE under contract no. DE-AC02-06CH11357. J.L.V. and P.A.C. gratefully acknowledge NSF grant CBET-1604971 and the facilities at the National Institute of Standards and Technology at Gaithersburg, M.D. as well as those at the John M. Cowley Center for High Resolution Electron Microscopy at Arizona State University. W.D.Y. acknowledges support under the Cooperative Research Agreement between the University of Maryland and the National Institute of Standards and Technology Physical Measurement Laboratory, award 70NANB14H209, through the University of Maryland. We also gratefully acknowledge stimulating discussions on supported catalyst materials with Dr. Christopher Karwacki that helped sharpen presentations made in the current work.

## Author contributions

Y.L., R.G.N., and A.I.F. conceived the project. Y.L., P.A.C., S.D.S., R.G.N., and A.I.F. designed the experiments. M.K. prepared the catalyst. J.L.V., L.Z., W.-C.D.Y., and P.A.C. were responsible for TEM experiments. Y.L. and M.K. tested the catalytic activity of the catalyst and performed DRIFTS measurements. Z.L. conducted AP-XPS experiments and analyzed AP-XPS data. Y.L., M.K., and J.H. conducted XAS experiments. Y.L. analyzed XAS data. Y.L., R.G.N., and A.I.F. co-wrote the manuscript with contributions from all authors. M.J.E. edited the manuscript.

## Competing interests

The authors declare no competing interests.
