## [Peer Review File · Nature Communications]

REVIEWER COMMENTS

Reviewer #1 (Remarks to the Author):

Authors present a detailed work on the WGS reaction on a Pt(1.85wt.)/CeO₂ catalyst. A battery of experimental techniques were employed to get insights into the dynamic behavior of the metal particles and the interface sites. Results are deeply analyzed and interpreted.

I find this work very interesting and I think it could help to the discussion about the role of metal-support interphase in the WGS reaction, if published.

Before, some aspects of the manuscript should be clarified:

1) IR experiments:

It is well known in the literature that the registered temperature in the “Temperature Reaction Chamber from Harrick Scientific Products” is VERY different from the real temperature of the catalyst bed. Please, indicate if you have inserted a thermocouple into the catalyst bed in order to correctly record the reaction temperature.

2)

Page 16 line 277

According to the XPS results,

Line 194 “...valence state of Pt averaged over the entire nanocluster, positively charged Pt²⁺ atoms (SI and Fig. S6), are most likely located at the interface with the Ce oxide support.”

If I understood well, there is a high concentration of Pt²⁺ atoms at the interface. Then, it is not clear to me what is the role assigned by the authors to the Pt²⁺ sites. Could you clarify this point in the text.

3)

Page 16 line 292

“Regenerative ‘redox’ mechanism requires that these changes (oxidation of ceria and perimeter Pt atoms is replaced by their reduction) reverse dynamically, consistent with the dynamic restructuring of the perimeter atoms observed by TEM.”

What is the meaning of “(oxidation of ceria and perimeter Pt atoms is replaced by their reduction)” ?

4)

The authors seems to described the redox mechanism for the WGS reaction according to the “metal assisted Mars van Krevelen” mechanism (see for instance: ACS Catal. 2018, 8, 7, 6513–6525).

I think the authors should clarify the discussion on this mechanism, where a reversible change of Ce⁺³⁻

to-Ce⁴⁺ should be operating. In this context, their results should be further discussed. Additionally, the authors consider that the presence of carbonaceous species are mere spectators. In this regard, it should be noted that the observation of apparently static signals under steady state conditions is not sufficient to dismiss an “associative mechanism”. Following the same reasoning, “almost static” CO signals are reported by the authors. Again, the authors must clarify this point.

Reviewer #2 (Remarks to the Author):

The communication by Li et al. addresses the structure of a ceria-supported Pt catalyst for the water gas shift (WGS) reaction. To gain insight into the nature and the dynamic formation of the active site in situ TEM, DRIFTS, XPS, and XAS were applied and a WGS reaction mechanism was proposed. As a major outcome of their study the authors claim perimeter PtO - O vacancy - Ce³⁺ sites to be the active site for the WGS reaction. The main criticism is related to this hypothesis, which is based on results of in situ experiments, which are not related to any conversion data and only part of which were performed at realistic pressure conditions. Thus their relevance for the WGS reaction remains unclear at this point. The manuscript may be suitable for publication in Nature Communications after the following points were carefully addressed by the authors:

1. As described above, activity data for the in situ experiments need to be added to show their relevance for the WGS reaction. Another issue concerns the very different in situ (cells) and pressure conditions. Comparability?
2. Figure 3: The observed peak profile of the DRIFTS data is quite complex suggesting several contributions. Did the authors perform a peak deconvolution? In particular, there appears to be an additional contribution at 2030-2040 cm⁻¹.
3. Figure 3: Did the authors observe DRIFTS features due to CO adsorbates on bare ceria? (DRIFTS data should be displayed over broader range, i.e., up to 2200 cm⁻¹.)
4. Figure 3: The basis of the assignment of the LF peaks is essential to the whole manuscript but remains rather indefinite: Which features (positions) are assigned to which type of CO species?
5. Figure 5: I am not convinced that the S/N of the Pt4f photoemission data at elevated temperatures (e.g. at 200 and 300°C) is sufficient for a detailed analysis of oxidation state changes. The authors are encouraged to show the XPS data in Figure S5 in a more transparent way (e.g. smaller symbols) to allow a better comparison of the changes. Besides, it is claimed that ‘PtO perimeter sites strongly associate with reduced ceria sites as indicated by DRIFTS and Ce3d AP-XPS data’. Please explain.
6. In the context of the proposed mechanism, the authors mention that the water reactant dissociates into hydroxyl and H species. Rather than forming hydroxyl groups the latter are assumed to reside in the proximity of oxygen vacancies. Is there any support for this? How is molecular hydrogen formed?

Minor:

7. For clarity, the authors are encouraged to display temperature dependent HF peak positions and use

the same scale in Figure 3(b)-(f).

8. In several passages of the manuscript the temperature unit is missing.

Reviewer #3 (Remarks to the Author):

The present work attempted in a well-thought and designed in situ/operando spectroscopic studies (HAADF/STEM, EXAFS, DRIFTS) to characterize for the first time at the atomic level the chemical structure/nature of the active sites of the water gas shift (WGS) reaction on a CeO₂-supported Pt catalytic system (mean Pt particle size of 1.7 nm). At the same time, the dynamics of Pt species (Pt(o) and Pt(n+)) evolution under WGS working reaction conditions was elucidated, for the first time to the knowledge of this reviewer. This fascinating work now presents in situ experimental proof for earlier DFT computational studies and advanced experimental kinetic studies reported in the literature for the same Pt/CeO₂ catalytic system, which suggested or provided strong evidence about the nature of active catalytic sites, namely, Pt sites at the perimeter of Pt-ceria interface with the participation of oxygen vacancies (Vo) of reducible ceria (Pt-Vo-Ce³⁺ sites).

This work confirms in the most convincing approach the importance of catalytic sites at the interface formed between a metal catalytic phase and support (supported metal catalyst), that even a rather small fraction of such sites could largely account for the experimentally observed reaction rate. This finding would definitely advance future design of supported metal catalysts on reducible supports (presence of Vo) for the optimization of the structure of such active sites per gram of catalyst basis. The manuscript is clearly written and the Conclusions are solid. The manuscript, however, can be strengthened if the present findings are more appropriately discussed with some published work regarding the same topic to be indicated below. Also, some missing experimental information necessary must be provided. A Minor Revision is suggested.

Comments:

1. An important review paper on the nature of active catalytic sites and adsorbed species in the WGS over supported Pt and Au catalysts, that covers specifically the use of advanced transient isotopic techniques, is missing (see Efstathiou, A.M., Elucidation of mechanistic and kinetic aspects of water-gas shift reaction on supported Pt and Au catalysts via transient isotopic techniques, *Catalysis*, 28, 175-236 (2016). This paper to be cited will provide to the readership the most important information on the mechanisms and active sites of WGS on the catalysts of interest in this work. Authors should introduce this review paper in Section "Main".

2. Kinetic rates of WGS on Pt/CeO₂ were expressed per length of the perimeter of Pt-CeO₂ interface (mols CO/cm/s) and correlated with a linear relationship with the Pt particle size for the first time (see Ref. [47]). This work addressed and discussed the importance of the structure of sites the present work elucidated via in situ/operando spectroscopic evidence. Therefore, it is very important for the readership to know that the results of the present work provide strong support and confirm earlier

indications on the likely structure of active sites of WGS on Pt/CeO₂ based on kinetic measurements. At the same time, the authors should provide some good discussion and their view based on the present results about the above-mentioned linear relationship reported (Ref. [47]). Is the dynamics of Pt active sites still operative for larger than 1.7 nm Pt particle sizes?

3. The importance of V_o could be further discussed in this paper based on experimental evidence and advanced kinetic experiments presented in the literature regarding the catalytic system CeO₂-doped supported Pt (La, Ti and Zr as dopants, see Petalidou K.C. et al., *Catalysis Today* 228, 183-193 (2014)).

4. Clarifications must be provided as to which conditions “post-reaction” catalyst is referred to (e.g. Fig. 1b, c).

5. Correct “Pt nanoparticle species” to “Pt nanoparticles”.

6. Fig. 3: Regarding Figs. 3b-3f, the same full scale for y-axis must be kept in order for the readership to read much easier the change in the signal intensity.

7. The DRIFTS observations (Fig. 3) must be better discussed. There is no experimental evidence from the present work or otherwise that a migration (surface diffusion) of CO adsorbed on HF Pt sites towards LF Pt sites occurs on increasing the reaction T in WGS. Simply, the effect of T on the surface coverage of CO for the Pt HF sites might be that arising from the reversible and exothermic character of CO adsorption. The increase of intensity of CO-s on LF Pt sites is small (bridged type CO-s), and since this overlaps with the linear type CO-s, without an accurate deconvolution this assignment is not convincing.

8. The experimental DRIFTS results shown in Fig. 4 find strong support by other earlier reported transient kinetic operando experiments (DRIFTS-Mass spectrometry) (see Ref. 47), which must be mentioned and briefly discussed.

9. Under the section “The catalytic activity at/near the perimeter Pto sites”, the authors discuss possible inactive, or spectator species formed in WGS. This is a very important topic that the authors must be careful what they are saying/writing. Ref. [47] and that mentioned in point 1 above, clearly discuss that only SSITKA-DRIFTS and other transient isotopic experiments are the appropriate ones to identify and discriminate active vs inactive species in WGS.

10. In Methods “DRIFTS”: It is very important that the authors confirm that background spectra obtained in He flow at given T do not differ from those obtained if H₂O/He were to be used. This is very important since water could have changed the IR signal absorbed by the solid and not related to intermediate species formed in the presence of water.

- END -

REVIEWER COMMENTS

Reviewer #1 (Remarks to the Author):

Authors present a detailed work on the WGS reaction on a Pt(1.85wt.)/CeO₂ catalyst. A battery of experimental techniques were employed to get insights into the dynamic behavior of the metal particles and the interface sites. Results are deeply analyzed and interpreted. I find this work very interesting and I think it could help to the discussion about the role of metal support interphase in the WGS reaction, if published. Before, some aspects of the manuscript should be clarified:

Reply: We are grateful to the reviewer for these helpful comments.

1) IR experiments:

It is well known in the literature that the registered temperature in the “Temperature Reaction Chamber from Harrick Scientific Products” is VERY different from the real temperature of the catalyst bed. Please, indicate if you have inserted a thermocouple into the catalyst bed in order to correctly record the reaction temperature.

Reply: We fully agree with the reviewer that the setup should be optimized for the Harrick cell to obtain the real temperature of the catalyst in the in-situ DRIFTS measurements. In our IR setup, the thermocouple is inserted into the catalyst bed. Furthermore, the trends observed across temperature for our DRIFTS measurements match the trends in XAS evaluation. The temperature-dependent DRIFTS spectra (Fig. S3) suggest Pt atom species in the as-prepared sample coarsen to metallic Pt nanoparticles at $T \geq 180$ °C to give the active catalyst. These metallic Pt nanoparticles remain the predominant species upon ramp-down to RT. A similar trend is also observed by XAS spectra (shown below; *added as Fig. S4 in the SI*): when $T < 200$ °C, with the increase of temperature, the catalyst grows from single atoms to nanoclusters. After being treated at 200 °C, 250 °C, and 300 °C, the XAS spectra are similar. For the XAS measurements, a flow reactor was used, and the thermocouple was inserted into the catalyst bed. Therefore, the similarity of the temperature-dependent structure evolution obtained from DRIFTS and XAS confirms that in our DRIFTS setup, the temperature was recorded correctly.

Figure S4. The comparison of (a) normalized XANES and (b) Fourier transformed $k^2\chi(k)$ EXAFS spectra of Pt L_3 edge of Pt/CeO₂ under WGS environment at RT (WGS_26°C), after 100°C WGS condition (WGS_100°C_26°C), after 150°C WGS condition (WGS_150°C_26°C), after 200°C WGS condition (WGS_200°C_26°C), after 250°C WGS condition (WGS_250°C_26°C) and after 300°C WGS condition (WGS_300°C_26°C). For reference, the spectra of Pt foil and α -PtO₂ were included. When $T < 200$ °C, with the increase of temperature, the catalyst grows from single atoms to nanoclusters. After being treated at 200 °C, 250 °C, and 300 °C, the XAS spectra are similar.

Accordingly, in Page 4,

“These results, together with those obtained from in situ diffuse reflectance infrared Fourier-Transform spectroscopy (DRIFTS) measurements (see DRIFTS details in the supporting information (SI) and Fig. S3), suggest that during the activating temperature ramp, single Pt atom species in the as-prepared sample coarsen to metallic Pt nanoparticles at $T \geq 180$ °C.”

was changed to

“These results, together with those obtained from in situ diffuse reflectance infrared Fourier-Transform spectroscopy (DRIFTS) measurements (see DRIFTS details in the supporting information (SI) and Fig. S3) and XAS measurements (Fig. S4), suggest that during the activating temperature ramp, single Pt atom species in the as-prepared sample coarsen to metallic Pt nanoparticles at $T \geq 180$ °C.”

2) Page 16 line 277

According to the XPS results,

Line 194 “...valence state of Pt averaged over the entire nanocluster, positively charged Pt²⁺ atoms (SI and Fig. S6), are most likely located at the interface with the Ce oxide support.”

If I understood well, there is a high concentration of Pt²⁺ atoms at the interface. Then, it is not clear to me what is the role assigned by the authors to the Pt²⁺ sites. Could you clarify this point in the text.

Reply: At low temperatures, when the catalyst is not active, the interface is dominated by Pt²⁺ atoms. As we have shown, about 22% of the interfacial Pt located at the perimeter sites are reduced to Pt⁰ at 300°C and these atoms are dynamic and active in the WGS reaction. Of the remaining interfacial atoms are Pt²⁺ and interact with the ceria support, thereby preventing further aggregation of small clusters. Furthermore, the additional space formed at the interface due to the elongated Pt²⁺-O distance is used to take in atomic H, which recombines into H₂ at working temperatures (discussed in the “Fluxional bond dynamics of Pt nanoclusters” Section).

In the revised manuscript, in Page 11, we added:

“The rest of the atoms at the interface remain in 2+ state and interact with the ceria support, thereby preventing further small cluster agglomeration.”

3)

Page 16 line 292

“Regenerative ‘redox’ mechanism requires that these changes (oxidation of ceria and perimeter Pt atoms is replaced by their reduction) reverse dynamically, consistent with the dynamic restructuring of the perimeter atoms observed by TEM.”

What is the meaning of “(oxidation of ceria and perimeter Pt atoms is replaced by their reduction)” ?

Reply: We are grateful to the reviewer for pointing out that our original statement was confusing. The original intent of that statement was to state that the regenerative ‘redox’ mechanism requires the dynamic oxidation state change of the active site (perimeter Pt⁰-O vacancy-Ce³⁺ site).

Accordingly, we made changes to the main text as follows.

In Page 15, the sentence:

“Regenerative ‘redox’ mechanism requires that these changes (oxidation of ceria and perimeter Pt atoms is replaced by their reduction) reverse dynamically, consistent with the dynamic restructuring of the perimeter atoms observed by TEM.”

was replaced by:

“Regenerative ‘redox’ mechanism requires oxidation state changes are dynamically reversible (reversible change of Pt⁰ to Pt²⁺ and Ce³⁺ to Ce⁴⁺). This is consistent with the dynamic restructuring of the perimeter atoms observed by TEM.”

4)

The authors seems to described the redox mechanism for the WGS reaction according to the “metal assisted Mars van Krevelen” mechanism (see for instance: ACS Catal. 2018, 8, 7, 6513–6525).

I think the authors should clarify the discussion on this mechanism, where a reversible change of Ce³⁺-to-Ce⁴⁺ should be operating. In this context, their results should be further discussed.

Additionally, the authors consider that the presence of carbonaceous species are mere spectators. In this regard, it should be noted that the observation of apparently static signals under steady state conditions is not a sufficient to dismiss an “associative mechanism”.

Following the same reasoning, “almost static” CO signals are reported by the authors. Again, the authors must clarify this point.

Reply: We appreciate the reviewer's comments and suggestions to help improve the manuscript. We highlighted the two main mechanistic pathways for the WGS reaction over ceria supported Pt nanocatalysts (described in the introduction) that have been proposed and remain the subject of debates: the regenerative (or redox) and associative mechanisms. Both proposed mechanisms involve the metal particles and the oxidic support as active sites. In the redox mechanism, CO adsorbed on metal sites reacts with an oxygen atom from the support to produce CO₂, and subsequently, water replenishes the generated oxygen vacancy (Applied Catalysis A: General, 215, 271-278, 2001). As the reviewer noticed, the redox mechanism for the WGS reaction over ceria supported catalysts is similar to the Mars van Krevelen mechanism for the CO oxidation reaction studied in ACS Catal. 2018, 8, 7, 6513–6525. Both involve two steps: transfer of oxygen from the support to the metal interface and re-oxidation of the support. The term “regenerative or redox mechanism” is more commonly used in studying the WGS reaction.

Following the reviewer's suggestion, we clarified the text by making the following changes:

In Page 15, the sentence:

“Regenerative ‘redox’ mechanism requires that these changes (oxidation of ceria and perimeter Pt atoms is replaced by their reduction) reverse dynamically, consistent with the dynamic restructuring of the perimeter atoms observed by TEM.”

was replaced by

“Regenerative ‘redox’ mechanism requires oxidation state changes are dynamically reversible (reversible change of Pt⁰ to Pt²⁺ and Ce³⁺ to Ce⁴⁺). This is consistent with the dynamic restructuring of the perimeter atoms observed by TEM.”

And in Page 16, the sentence:

“The higher flexibility of the perimeter atoms has also been observed in ceria supported Au nanoparticles under CO oxidation, a reaction that most likely proceeds via oxygen vacancy exchange.”

was changed to

“The higher mobility of the perimeter atoms has also been observed in ceria supported Au nanoparticles under CO oxidation, a reaction that most likely proceeds via a metal-assisted Mars–van Krevelen mechanism with oxygen vacancy exchange involved.”

While our work demonstrates that the reaction most likely progresses via the regenerative ‘redox’ mechanism at the perimeter Pt⁰–O vacancy–Ce³⁺ sites, we fully agree with the reviewer that we are not able to completely dismiss an “associative mechanism” and determine the active/spectator species basing on the static signals under steady state conditions only. In order to further decrease the ambiguity between active and spectator species in the WGS reaction, in addition to the experiments that we have already performed, advanced transient isotopic experiments could have been useful, and we modified the manuscript to address that point.

In Page 15, the sentence:

“Though carbon-containing intermediates (formate, carbonate, carboxylic acid) are present (Fig. S8), they are more likely spectators since they are present even at low temperatures when the catalyst shows no activity. Furthermore, their corresponding DRIFTS bands show almost no changes upon the change of external condition from CO to WGS. We therefore believe an alternative model based on a reaction proceeding via a regenerative ‘redox’ mechanism to be the more likely one followed.”

was changed to

“Though the associative mechanism may also play role (Fig. S10), the perimeter Pt⁰-O vacancy-Ce³⁺ sites are more likely associated with a regenerative ‘redox’ mechanism. By using steady-state isotopic transient kinetic analysis (coupled with DRIFTS and mass spectrometry), Kalamaras et al. studied Pt/ceria catalysts of different sizes (1.3 to 8.0 nm). These results also suggest that at 300 °C, the WGS reaction proceeds largely via the ‘redox’ mechanism with the reactive site located along the periphery of the Pt-CeO₂ interface (the specific rate based on the length of periphery of the Pt-ceria interface increases linearly with respect to particle size).”

In the supporting information,

“Figure S8. The DRIFTS spectra at different temperatures (ramp-down process) under WGS and CO conditions for Pt/CeO₂ catalyst. The assignments to the peaks are based on the literature⁶⁻⁹.”

was changed to

“Figure S10. The DRIFTS spectra at different temperatures (ramp-down process) under WGS and CO conditions for Pt/CeO₂ catalyst. The assignments to the peaks are based on the literature⁶⁻⁹. We note here that the existence of carbon-containing intermediates (formate, carbonate, carboxylic acid) suggests that the associative mechanism may also play role. However, to identify or discriminate active/spectator species in the WGS reaction, advanced transient isotopic experiments are needed.”

Reviewer #2 (Remarks to the Author):

The communication by Li et al. addresses the structure of a ceria-supported Pt catalyst for the water gas shift (WGS) reaction. To gain insight into the nature and the dynamic formation of the active site in situ TEM, DRIFTS, XPS, and XAS were applied and a WGS reaction mechanism was proposed. As a major outcome of their study the authors claim perimeter Pt⁰ - O vacancy - Ce³⁺ sites to be the active site for the WGS reaction. The main criticism is related to this hypothesis, which is based on results of in situ experiments, which are not related to any conversion data and only part of which were performed at realistic pressure conditions. Thus their relevance for the WGS reaction remains unclear at this point. The manuscript may be suitable for publication in Nature Communications after the following points were carefully addressed by the authors:

1. As described above, activity data for the in situ experiments need to be added to show their relevance for the WGS reaction. Another issue concerns the very different in situ (cells) and pressure conditions. Comparability?

Reply: We thank the reviewer for their valuable comments and questions. We performed additional experiments to better link catalyst characterization with conversion data. We agree with the reviewer that, in order to correlate results obtained from different measurements, it is necessary to make sure that the catalyst behaves similarly in different *in-situ* measurements. For some of our experiments, we have already conducted measurements as the reviewer recommended. Specifically, for the activity results (Figure 1a), our Pt/ceria catalyst was tested in a Clausen reactor (J. Appl. Crystallogr. **41**, 822-824 (2008)). For the in-situ XAS measurements, the same Clausen reactor and the same *in-situ* setup were used, hence, the structural results and the activity were measured for the same state of the catalyst and the reaction in the both experiments, and therefore could be directly correlated. We made an additional clarification in the manuscript emphasizing the need for such a correlative measurement for linking structure and activity measurements, as the reviewer pointed out. For the DRIFTS measurement, a Praying Mantis High Temperature Reaction Chamber (Harrick Scientific Products) was used. To confirm that the catalytic behavior of the catalyst is the same in the Clausen cell and in the IR cell, we performed new experiments, in which we tested the catalyst using the IR cell and collected the MS data, which are in a good agreement with the MS data reported in the manuscript when using the Clausen cell. This new MS data is shown below and now included in the SI (Figure S11).

Accordingly, in “Methods” Section, in Page 19, we added

“Using this reactor, the catalytic behavior of the catalyst was also tested and found to be very similar to that obtained using the Clausen reactor (Fig. S11).”

We note that MS can validate the correlation between results obtained using different *in situ* techniques with varying reaction cells (such as those optimized for XAFS and TEM, as used by us and by others: see, e.g., Y. Li, et al, Nature Commun. 2015, 6, 7583), and furthermore that this provides an operando approach, when multiple techniques cannot be combined in the same experiment.

Figure S11. Activity test using the DRIFTS reactor: the H₂ and CO₂ productions at different temperatures in the ramp up and down processes. Similar trend was observed using the Clausen cell (Fig. 1(a)).

Applying such a correlative investigation to the studies by AP-XPS and ETEM is not feasible in this work, due to the small mass of catalyst sample used in both techniques (which poses difficulty in detecting conversion) and due to the scattering of photoelectrons/electrons by gas molecules (which poses difficulty in performing XPS and ETEM at high pressures without sacrificing data quality). For AP-XPS and ETEM measurements, at the state of the art in such methodologies, although the detection of conversion is difficult, the agreement of their results with those obtained from the activity test, XAS, and DRIFTS suggests that the catalyst behaves similarly in all those experiments. For example, the AP-XPS results show the concentration of Ce³⁺ increases starting at about 200°C and up to 300°C (Fig. 5b). The DRIFTS studies of hydroxyl species on the surface of the ceria support (Fig. 4) also suggest the reduction of the ceria support starting from 200°C and up to 300°C. The AP-XPS results show the concentration increase of Pt⁰ starting at about 200°C (Fig. 5a). The *in-situ* XANES data also suggest the reduction of the Pt nanocatalyst from 200°C to 300°C. The *in-situ* TEM image shows a distortion in the interplanar spacing (plane bending) (Fig. 2(b2)) under the WGS condition. The *in-situ* EXAFS data also show the change of Pt-O and Pt-Pt bond distance under reaction conditions (Fig. 6). The *in-situ* TEM measurements suggest that the particle size of the active Pt nanocluster is about 1.7 nm, agree with the estimation obtained basing on EXAFS data analysis (discussed in the “Fluxional bond dynamics of Pt nanoclusters” Section). As such, although in some experiments we directly correlated the catalyst structure changes with activity, and in others we infer the activity indirectly, the consistency of the structural behaviors suggest that the catalyst behaves similarly in different measurements and show the importance of multimodal approach.

2. Figure 3: The observed peak profile of the DRIFTS data is quite complex suggesting several

contributions. Did the authors perform a peak deconvolution? In particular, there appears to be an additional contribution at 2030-2040 cm^{-1} .

Reply: We thank the reviewer for these questions. Indeed, several peaks could be observed in the DRIFTS data which suggests Pt atoms with different bonding environments exist on the surface of the nanocatalyst. Guided by the *in-situ* TEM results that the perimeter atoms are unique, we aimed to get more information about the perimeter Pt sites from DRIFTS measurements. Therefore, the peaks in Fig. 3 are grouped into two types: one is LF peaks ($2020\text{-}1900\text{ cm}^{-1}$) associated with CO bonded to Pt atoms in proximal contact with reduced Ce^{3+} sites along the metal-support interface; the other is peaks with the wavenumber higher than 2020 cm^{-1} and they are associated with CO on surface Pt atoms with different coordination environment. For instance, we observed peaks between $2072\text{-}2054\text{ cm}^{-1}$ and they are assigned to linearly adsorbed CO on terrace or step Pt sites. From Fig. 3a, the DRIFTS spectra are smooth in the region of $2050\text{-}2020\text{ cm}^{-1}$ and no band features could be observed between $2040\text{-}2030\text{ cm}^{-1}$. If they do exist, in this region, the peaks are assigned to CO adsorbed on low coordinated Pt^0 surface atoms (Phys. Chem. Chem. Phys., 2005, 7, 187-194). Such grouping better shows the change of CO adsorption on perimeter Pt^0 atoms and other surface Pt^0 atoms with temperature (Fig. 3a) and with the change of gas condition (Fig. 3b-3f). We therefore did not do the peak deconvolution to identify the change of each peak.

3. Figure 3: Did the authors observe DRIFTS features due to CO adsorbates on bare ceria? (DRIFTS data should be displayed over broader range, i.e., up to 2200 cm^{-1} .)

Reply: For CO adsorbed on Ce^{4+} or Ce^{3+} , the bands reside between $2120\text{-}2180\text{ cm}^{-1}$ (J. Catal. 2006, 237, 1-16). As one can see below, no signal could be observed in this region in our data. Because of this and because we want to better show the changes of LF peaks, in our Fig. 3, we used narrower range: $2130\text{-}1860\text{ cm}^{-1}$. The DRIFTS spectra (under CO and WGS conditions) with broad range ($4000\text{-}800\text{ cm}^{-1}$) were already shown in Fig. S10.

Figure caption: The temperature dependent CO – DRIFTS spectra of Pt/CeO₂ catalyst collected in the ramp-down process.

4. Figure 3: The basis of the assignment of the LF peaks is essential to the whole manuscript but remains rather indefinite: Which features (positions) are assigned to which type of CO species?

Reply: The band centered at ~2016 cm⁻¹ is assigned to CO linearly adsorbed on perimeter Pt sites while the band centered at ~1956 cm⁻¹ is assigned to a bridged CO adsorption at the metal support interface. Due to the word limits of the main text, the band assignments for LF peaks were listed in the caption of Fig. 3.

5. Figure 5: I am not convinced that the S/N of the Pt4f photoemission data at elevated temperatures (e.g. at 200 and 300°C) is sufficient for a detailed analysis of oxidation state changes. The authors are encouraged to show the XPS data in Figure S5 in a more transparent way (e.g. smaller symbols) to allow a better comparison of the changes. Besides, it is claimed that ‘Pt⁰ perimeter sites strongly associate with reduced ceria sites as indicated by DRIFTS and Ce3d AP-XPS data’. Please explain.

Reply: We thank the reviewer for the questions and suggestions. Due to the relatively low concentration of Pt in our Pt/ceria sample and the small amount of the sample used in the AP-XPS measurement, the signal/noise level is low for Pt. However, the change of oxidation state of Pt atoms could still be observed in the raw data. Following the reviewer’s suggestion, we updated Figures S5 (now Figure S7 in the updated SI). In the figure (also shown below), we used a smaller symbol for the raw data. As shown in (a), at 300 °C, the peak at about 72 eV in the raw data is higher than the one at about 75 eV. At 200 °C, these two peaks are comparable. In addition to that, the peaks at 200 °C are broader than those at 300 °C. These observations suggest that at 300 °C the concentration of Pt⁰ (green) is higher while at 200 °C the concentration of Pt²⁺ (blue) is higher. Such trend observed in the raw data agrees with the trend summarized in Fig. 5(a).

Figure S7. The (a) Pt 4f and (b) Ce 3d region XPS spectra (black) and corresponding fitting spectra (red) for the ceria supported Pt cluster under the WGS reaction conditions at different temperatures. For Pt 4f XPS spectra, the data were fitted by Pt^{2+} (blue) and Pt^0 (green) components. For Pt^{2+} , the experimentally observed binding energy of the Pt-4f_{7/2} core level is 72.8 eV and for Pt^0 , is 71.5 eV. For Ce 3d XPS spectra, the data were fitted by Ce^{4+} (blue) and Ce^{3+} (green) components. For Ce^{4+} , the experimentally observed binding energy of the Ce-3d_{5/2} core level is 882.6 eV and of the Ce-3d_{3/2} core level is 901.2 eV. For Ce^{3+} , the experimentally observed binding energy of the Ce-3d_{5/2} core level is 880.6 eV and of the Ce-3d_{3/2} core level is 899.2 eV.

According to the XPS data, at high temperatures, some perimeter Pt atoms are reduced from 2+ to the zerovalent state. These high temperature Pt^0 perimeter sites strongly associate with reduced ceria sites because we also observed the perimeter Pt^0 -O vacancy- Ce^{3+} sites from the DRIFTS spectra only at high temperatures (Fig. 3). In addition to that, the reduction of Ce^{4+} was detected at high temperatures under the WGS condition by AP-XPS (Fig. 5(b)). Based on these observations, we claimed that “These high temperature Pt^0 perimeter sites strongly associate with reduced ceria sites as indicated by DRIFTS and Ce 3d AP-XPS data.”

To make the statement clearer, the sentence in Page 11:

“These high temperature Pt^0 perimeter sites strongly associate with reduced ceria sites as indicated by DRIFTS and Ce 3d AP-XPS data. Under the WGS reaction, the concentration of Ce^{3+} is 20% at 300 °C, which is 5% higher than that at RT.”

was changed to

“These high temperature Pt⁰ perimeter sites strongly associate with reduced ceria sites as indicated by DRIFTS (the emergence of high temperature perimeter Pt⁰-O vacancy-Ce³⁺ sites) and Ce 3d AP-XPS data. Under the WGS reaction, the concentration of Ce³⁺ is 20% at 300 °C, which is 5% higher than that at RT.”

6. In the context of the proposed mechanism, the authors mention that the water reactant dissociates into hydroxyl and H species. Rather than forming hydroxyl groups the latter are assumed to reside in the proximity of oxygen vacancies. Is there any support for this? How is molecular hydrogen formed?

Reply: We thank the reviewer for these questions, and we have clarified these points accordingly in the revised manuscript. First, yes, we believe that, rather than forming hydroxyl groups, the hydrogen atoms are present as neutral atomic species. From our EXAFS results, the long Pt-O bond was detected and ascribed to the presence of a hydrogen atom residing at the interface between the metal atoms and the support. The hydrogen may present as protonic hydrogen (as from support hydroxyl groups) or neutral atomic species. In this work, we suggest that the following observations are more consistent with the latter interpretation: 1) liberation of interfacial hydrogen at high temperatures was not accompanied by oxidation of the platinum (the AP-XPS and XANES data suggest the reduction of the Pt nanocluster) as would be expected if protonic hydrogen was reduced to form H₂. 2) Similarly, interfacial Pt atoms are expected to be reduced if they are interacting with protonic hydrogen. However, in this work, the interfacial Pt atoms remain in the 2+ state.

Second, the long Pt-O bond reflects the metal-support interaction. As discussed above, the atomic hydrogen atoms stay at the interface between the Pt nanocluster and the ceria support.

In Page 15, the statements of *“The reactant H₂O, via dipole-dipole interaction with CO, affect the CO-Pt bonding, and further likely dissociates into OH and atomic H proximal to the oxygen vacancy sites on the surface of the ceria support; the hydroxyl groups bond to the surface of the ceria support, leaving atomic H species at the interface of the Pt nanocluster and the support.”*

were changed to

“The reactant H₂O, via dipole-dipole interaction with CO, affects the CO-Pt bonding, and further likely dissociates into hydroxyl and atomic H: the hydroxyl groups bond to the surface of the ceria support, leaving atomic H species at the interface of the Pt nanocluster and the support.”

According to our combined results, the adsorbed H atoms recombine into H₂ at working temperatures and such recombination occurs with the appearance of active oxygen vacancies (near perimeter Pt⁰ sites) and they are suggested to act as specific sites for H₂ activation according to previous work (J. Phys. Chem. B 2004, 108, 18509-18519).

Minor:

7. For clarity, the authors are encouraged to display temperature dependent HF peak positions and use the same scale in Figure 3(b)-(f).

Reply: We thank the reviewer for the suggestion. As suggested, we modified Figs. 3(b)-(f) and now they have the same y scale. Figures 3b-3f are used to show the changes of HF and LF peaks at different temperatures under the WGS condition. Instead of labelling the position of HF peaks in figures 3b-3f, we prepared a new Fig. S6 in the SI. In Fig. S6, the positions of HF peaks of spectra collected at different temperatures in CO and WGS conditions were labeled for readers references. Accordingly, in Page 9, at the end of the caption of Fig. 3, we added “*The positions of HF peaks were labeled in Fig. S6*”.

8. In several passages of the manuscript the temperature unit is missing.

Reply: Thank you for pointing out the errors created during Word-PDF conversion. We fixed all of these in the revised version of the manuscript.

Reviewer #3 (Remarks to the Author):

The present work attempted in a well-thought and designed in situ/operando spectroscopic studies (HAADF/STEM, EXAFS, DRIFTS) to characterize for the first time at the atomic level the chemical structure/nature of the active sites of the water gas shift (WGS) reaction on a CeO₂-supported Pt catalytic system (mean Pt particle size of 1.7 nm). At the same time, the dynamics of Pt species (Pt(o) and Pt(n+)) evolution under WGS working reaction conditions was elucidated, for the first time to the knowledge of this reviewer. This fascinating work now presents in situ experimental proof for earlier DFT computational studies and advanced experimental kinetic studies reported in the literature for the same Pt/CeO₂ catalytic system, which suggested or provided strong evidence about the nature of active catalytic sites, namely, Pt sites at the perimeter of Pt-ceria interface with the participation of oxygen vacancies (Vo) of reducible ceria (Pt-Vo-Ce³⁺ sites).

This work confirms in the most convincing approach the importance of catalytic sites at the interface formed between a metal catalytic phase and support (supported metal catalyst), that even a rather small fraction of such sites could largely account for the experimentally observed reaction rate. This finding would definitely advance future design of supported metal catalysts on reducible supports (presence of Vo) for the optimization of the structure of such active sites per gram of catalyst basis.

The manuscript is clearly written and the Conclusions are solid. The manuscript, however, can be strengthened if the present findings are more appropriately discussed with some published work regarding the same topic to be indicated below. Also, some missing experimental information necessary must be provided. A Minor Revision is suggested.

Reply: We are grateful to the reviewer’s praise of our work.

Comments:

1. An important review paper on the nature of active catalytic sites and adsorbed species in the WGS over supported Pt and Au catalysts, that covers specifically the use of advanced transient isotopic techniques, is missing (see Efstathiou, A.M., Elucidation of mechanistic and kinetic aspects of water-gas shift reaction on supported Pt and Au catalysts via transient isotopic techniques, *Catalysis*, 28, 175-236 (2016)). This paper to be cited will provide to the readership the most important information on the mechanisms and active sites of WGS on the catalysts of interest in this work. Authors should introduce this review paper in Section “Main”.

Reply: We agree with these suggestions. The manuscript was modified accordingly.

In Section “Introduction” (previously Section “Main”), in Page 3, the sentence:

“The mechanistic/kinetic studies, on the other hand, suggest that supports, especially oxygen vacancies of the supports, play an essential role in the activity of the catalyst because water dissociation involved in two main mechanisms (the regenerative/redox and the associative mechanism) is believed to occur at oxygen vacancy sites.”

was changed to

“Mechanistic/kinetic studies, based on advanced transient isotopic techniques, on the other hand, suggest that catalyst support provides active sites within a region (reactive zone) around the Pt nanoparticles. Supports, especially oxygen vacancies of the supports, play an essential role in the activity of the catalyst because water dissociation, involved in two main mechanisms (the regenerative/redox and the associative mechanisms), is believed to occur at oxygen vacancy sites.”.

The reference “Efstathiou, A. M. *Catalysis: Elucidation of mechanistic and kinetic aspects of water–gas shift reaction on supported Pt and Au catalysts via transient isotopic techniques* Ch. 7, **28**, 175-236 (The Royal Society of Chemistry, 2016)” is now in the reference list (Reference 10).

2. Kinetic rates of WGS on Pt/CeO₂ were expressed per length of the perimeter of Pt-CeO₂ interface (mols CO/cm/s) and correlated with a linear relationship with the Pt particle size for the first time (see Ref. [47]). This work addressed and discussed the importance of the structure of sites the present work elucidated via in situ/operando spectroscopic evidence. Therefore, it is very important for the readership to know that the results of the present work provide strong support and confirm earlier indications on the likely structure of active sites of WGS on Pt/CeO₂ based on kinetic measurements. At the same time, the authors should provide some good discussion and their view based on the present results about the above-mentioned linear relationship reported (Ref. [47]). Is the dynamics of Pt active sites still operative for larger than 1.7 nm Pt particle sizes?

Reply: As suggested, we expanded the discussion on the results obtained in this work and those obtained in previous work by Efstathiou et al (*J. Catal.* **279**, 287-300 (2011)).

In Page 15, the sentence:

“Though carbon-containing intermediates (formate, carbonate, carboxylic acid) are present (Fig. S8), they are more likely spectators since they are present even at low temperatures when the

catalyst shows no activity. Furthermore, their corresponding DRIFTS bands show almost no changes upon the change of external condition from CO to WGS. We therefore believe an alternative model based on a reaction proceeding via a regenerative 'redox' mechanism to be the more likely one followed."

was changed to

"Though the associative mechanism may also play role (Fig. S10), the perimeter Pt⁰-O vacancy-Ce³⁺ sites are more likely associated with a regenerative 'redox' mechanism. By using steady-state isotopic transient kinetic analysis (coupled with DRIFTS and mass spectrometry), Kalamaras et al. studied Pt/ceria catalysts of different sizes (1.3 to 8.0 nm). These results also suggest that at 300 °C, the WGS reaction proceeds largely via the 'redox' mechanism with the reactive site located along the periphery of the Pt-CeO₂ interface (the specific rate based on the length of periphery of the Pt-ceria interface increases linearly with respect to particle size)."

By combining the results obtained by Efstathiou et al (*J. Catal.* **279**, 287-300 (2011)) and the those obtained in this work (which highlights the fluxionality of the perimeter Pt⁰-O vacancy-Ce³⁺ site as the key activity descriptor), we think the dynamics of Pt active sites is still operative for larger than 1.7 nm Pt particle sizes. We would like to confirm it by studying Pt/ceria catalysts with larger size in the future work.

3. The importance of Vo could be further discussed in this paper based on experimental evidence and advanced kinetic experiments presented in the literature regarding the catalytic system CeO₂-doped supported Pt (La, Ti and Zr as dopants, see Petalidou K.C. et al., *Catalysis Today* **228**, 183-193 (2014)).

Reply: As suggested, in Page 16, we added: *"Experimentally, to improve the reactivity of Pt/ceria catalysts for the WGS reaction, the dopants should be able to increase the concentration or improve the activity of oxygen vacancies that involved in the perimeter Pt⁰-O vacancy-Ce³⁺ sites."* Also, the paper (*Catal. Today* **228**, 183-193 (2014)) was cited in the updated manuscript as reference 53.

4. Clarifications must be provided as to which conditions "post-reaction" catalyst is referred to (e.g. Fig. 1b, c).

Reply: Fig. 1b is the STEM annular dark field (ADF) image of the as-prepared Pt/ceria catalysts. Fig. 1c is the STEM-ADF image of the used Pt/ceria catalyst. The reacted/used catalyst is referred to the catalyst that collected after the activity test (Fig. 1a).

As the reviewer requested, we clarified this as follows:

In Page 4, the sentence:

"To understand the effect of WGS reaction conditions upon the structure of the catalyst, the nature of the as-prepared and reacted Pt/CeO₂ were evaluated using scanning transmission

electron microscopy (STEM) and X-ray absorption spectroscopy (XAS). HAADF-STEM images of the as-prepared and post-reaction catalyst are shown in Fig. 1(b) and Fig. 1(c), respectively.”

was changed to

“To understand the effect of WGS reaction conditions upon the structure of the catalyst, the nature of the as-prepared and reacted (collected after the activity test) Pt/CeO₂ were evaluated using scanning transmission electron microscopy (STEM) and X-ray absorption spectroscopy (XAS). HAADF-STEM images of the as-prepared and reacted catalyst are shown in Fig. 1(b) and Fig. 1(c), respectively.”

For Fig. 1c, the caption was changed from “*The STEM-ADF image of the used Pt/ceria catalyst*” to “*The STEM-ADF image of the reacted Pt/ceria catalyst*”.

5. Correct “Pt nanoparticle species” to “Pt nanoparticles”.

Reply: As the reviewer suggested, in Page 5, we replaced “*Pt nanoparticle species*” with “*Pt nanoparticles*”.

6. Fig. 3: Regarding Figs. 3b-3f, the same full scale for y-axis must be kept in order for the readership to read much easier the change in the signal intensity.

Reply: Figures 3b-3f were modified as suggested.

7. The DRIFTS observations (Fig. 3) must be better discussed. There is no experimental evidence from the present work or otherwise that a migration (surface diffusion) of CO adsorbed on HF Pt sites towards LF Pt sites occurs on increasing the reaction T in WGS. Simply, the effect of T on the surface coverage of CO for the Pt HF sites might be that arising from the reversible and exothermic character of CO adsorption. The increase of intensity of CO-s on LF Pt sites is small (bridged type CO-s), and since this overlaps with the linear type CO-s, without an accurate deconvolution this assignment is not convincing.

Reply: We would like to clarify that the decrease of HF peaks and the increase of LF peaks are observed when we change the gas condition from CO to WGS at the same temperature when the temperature is high (Figs. 3b-3f). Therefore, the decrease of HF peaks and the increase of LF peaks observed are not due to the temperature effects on CO adsorption but due to the change of gas environment, specifically the introduction of water to CO. The decrease of HF peaks and the increase of LF peaks occur simultaneously suggest the redistribution/migration of CO on the surface of Pt nanoparticles.

As labeled in Figure 3a, there are two LF peaks: one at about 2016 cm⁻¹ and another at about 1956 cm⁻¹. As the reviewer mentioned, LF peaks may overlap. Even they overlap, the increase of them could still suggest that there are more CO adsorbed on the low coordinated perimeter Pt active sites because both LF peaks are associated with CO bonded to low coordinated Pt atoms in

proximal contact with reduced Ce^{3+} sites along the metal–support interface ($\text{Pt}^0\text{–O}$ vacancy– Ce^{3+}). In Figs. 3b-3f, we did not label the change of the 2016 cm^{-1} because it may overlap with HF peaks. The decrease of HF peaks and the increase of LF peaks occur not only at $300\text{ }^\circ\text{C}$ (at which the change is relatively small) but also at $180\text{ }^\circ\text{C}$ and $240\text{ }^\circ\text{C}$ when the catalyst is active. At $180\text{ }^\circ\text{C}$, the decrease of HF peaks and the increase of LF peaks are most significant, and with the increase of temperature, the changes become smaller. We think that could be related to the temperature effect which results in the low coverage of CO at higher temperatures.

To avoid confusion, in Page 8, the sentence

“Upon further addition of H_2O to CO at 180°C , 240°C and 300°C , the intensities of the HF peaks decrease while those of the LF ones increase, an indication of the migration of some CO adsorbates from high to low coordination perimeter Pt sites.”

was changed to

“Upon further addition of H_2O to CO at 180°C , 240°C and 300°C , the intensities of the HF peaks decrease while those of the LF peaks increase, an indication of the migration of some CO adsorbates from high to low coordination perimeter Pt sites due to the change of gas condition from CO to WGS. Such changes seem stronger with the decrease of the temperature from $300\text{ }^\circ\text{C}$ to $180\text{ }^\circ\text{C}$ which could be due to the temperature effect on the CO coverage.”

8. The experimental DRIFTS results shown in Fig. 4 find strong support by other earlier reported transient kinetic operando experiments (DRIFTS-Mass spectrometry) (see Ref. 47), which must be mentioned and briefly discussed.

Reply: As suggested, in Page 9, we added: *“Previous reports also suggest that the active –OH groups are classified as type-II –OH groups formed on partially reduced ceria and reside within a narrow zone around the periphery of Pt–ceria interface.”*

9. Under the section “The catalytic activity at/near the perimeter Pto sites”, the authors discuss possible inactive, or spectator species formed in WGS. This is a very important topic that the authors must be careful what they are saying/writing. Ref. [47] and that mentioned in point 1 above, clearly discuss that only SSITKA-DRIFTS and other transient isotopic experiments are the appropriate ones to identify and discriminate active vs inactive species in WGS.

Reply: We completely agree with the reviewer that in the current work, we could not identify or discriminate active/spectator species in the WGS reaction. And to do that, advanced transient isotopic experiments should be performed.

In Page 15, the sentence:

“Though carbon-containing intermediates (formate, carbonate, carboxylic acid) are present (Fig. S8), they are more likely spectators since they are present even at low temperatures when the catalyst shows no activity. Furthermore, their corresponding DRIFTS bands show almost no

changes upon the change of external condition from CO to WGS. We therefore believe an alternative model based on a reaction proceeding via a regenerative ‘redox’ mechanism to be the more likely one followed.”

was changed to

“Though the associative mechanism may also play role (Fig. S10), the perimeter Pt^0 -O vacancy-Ce³⁺ sites are more likely associated with a regenerative ‘redox’ mechanism. By using steady-state isotopic transient kinetic analysis (coupled with DRIFTS and mass spectrometry), Kalamaras et al. studied Pt/ceria catalysts of different sizes (1.3 to 8.0 nm). These results also suggest that at 300 °C, the WGS reaction proceeds largely via the ‘redox’ mechanism with the reactive site located along the periphery of the Pt-CeO₂ interface (the specific rate based on the length of periphery of the Pt-ceria interface increases linearly with respect to particle size).”

In the supporting information,

“Figure S8. The DRIFTS spectra at different temperatures (ramp-down process) under WGS and CO conditions for Pt/CeO₂ catalyst. The assignments to the peaks are based on the literature⁶⁻⁹.”

was changed to

“Figure S10. The DRIFTS spectra at different temperatures (ramp-down process) under WGS and CO conditions for Pt/CeO₂ catalyst. The assignments to the peaks are based on the literature⁶⁻⁹. We note here that the existence of carbon-containing intermediates (formate, carbonate, carboxylic acid) suggests that the associative mechanism may also play role. However, to identify or discriminate active/spectator species in the WGS reaction, advanced transient isotopic experiments are needed.”

10. In Methods “DRIFTS”: It is very important that the authors confirm that background spectra obtained in He flow at given T do not differ from those obtained if H₂O/He were to be used. This is very important since water could have changed the IR signal absorbed by the solid and not related to intermediate species formed in the presence of water.

Reply: We agree with the reviewer that water in the system could affect the IR signal. We checked the background spectra collected in H₂O/He which confirm that the intermediate species mentioned in the manuscript are not due to water.

Additional changes in the manuscript and SI:

In Fig. 1a, the unit notation in the y axis label was modified.

REVIEWERS' COMMENTS

Reviewer #1 (Remarks to the Author):

Comments made in my original revision of the manuscript were well addressed by the Authors. Therefore, I recommend the publication of this manuscript.

Reviewer #2 (Remarks to the Author):

The authors have carefully addressed the questions raised in my report and have modified the manuscript accordingly. I therefore recommend publication of the revised version.

Reviewer #3 (Remarks to the Author):

The authors have very carefully addressed all comments of Reviewer No. 3 in a very satisfactory manner.

The revised Ms can now be accepted for publication in Nature Communications.

REVIEWER COMMENTS

Reviewer #1 (Remarks to the Author):

Comments made in my original revision of the manuscript were well addressed by the Authors. Therefore, I recommend the publication of this manuscript.

Reply: Thank you.

Reviewer #2 (Remarks to the Author):

The authors have carefully addressed the questions raised in my report and have modified the manuscript accordingly. I therefore recommend publication of the revised version.

Reply: Thank you.

Reviewer #3 (Remarks to the Author):

The authors have very carefully addressed all comments of Reviewer No. 3 in a very satisfactory manner.

The revised Ms can now be accepted for publication in Nature Communications.

Reply: Thank you.